# Consensus Statements among European Sleep Surgery Experts on Snoring and Obstructive Sleep Apnea: Part 2 Decision-Making in Surgical Management and Peri-Operative Considerations

**DOI:** 10.3390/jcm13072083

**Published:** 2024-04-03

**Authors:** Ewa Olszewska, Andrea De Vito, Carlos O’Connor-Reina, Clemens Heiser, Peter Baptista, Bhik Kotecha, Olivier Vanderveken, Claudio Vicini

**Affiliations:** 1Department of Otolaryngology, Sleep Apnea Surgery Center, Medical University of Bialystok, 15-276 Bialystok, Poland; 2Department of Surgery, Morgagni-Pierantoni Hospital, Health Local Agency of Romagna, 47121 Forlì, Italy; 3Hospitales Quironsalud Marbella, 29603 Malaga, Spain; carlos.oconnor@quironsalud.es; 4Faculty of Medicine and Health Sciences, University of Antwerp, 2000 Antwerp, Belgium; hno@heiser-online.com (C.H.); olivier.vanderveken@uza.be (O.V.); 5Department of Otorhinolaryngology/Head and Neck Surgery, Klinikum Rechts der Isar, Technical University of Munich, 80333 Munich, Germany; 6Clinica Universidad da Navarra, Departmento de Orl, 31008 Pamplona, Spain; peterbaptista@gmail.com; 7Nuffield Health Brentwood, Essex, Brentwood CM15 8EH, UK; bhikkot@aol.com; 8UME Health, 17 Harley Street, London W1G 9QH, UK; 9Department of Otorhinolaryngology, Head and Neck Surgery, Antwerp University Hospital, 2650 Antwerp, Belgium; 10GVM Care & Research ENT Consultant, GVM Primus Medica Center, GVM San Pier Damiano Hospital, 48018 Faenza, Italy; claudio@claudiovicini.com

**Keywords:** snoring, sleep apnea, obstructive sleep apnea, consensus, decision making, treatment, surgical management, peri-operative considerations

## Abstract

**Background**: Reaching consensus on decision-making in surgical management and peri-operative considerations regarding snoring and obstructive sleep apnea (OSA) among sleep surgeons is critical in the management of patients with such conditions, where there is a large degree of variability. **Methods**: A set of statements was developed based on the literature and circulated among eight panel members of European experts, utilizing the Delphi method. Responses were provided as agree and disagree on each statement, and the comments were used to assess the level of consensus and develop a revised version. The new version, with the level of consensus and anonymized comments, was sent to each panel member as the second round. This was repeated for a total of five rounds. **Results**: The final set included a total of 71 statements: 29 stand-alone and 11 with 42 sub-statements. On the 33 statements regarding decision-making in surgical management, there was 60.6%, 27.3%, and 6.1% consensus among all eight, seven, and six panelists, respectively. On the 38 statements regarding the peri-operative considerations, there was 55.3%, 18.4%, and 15.8% consensus among all eight, seven, and six panelists, respectively. **Conclusions**: These results indicate the need for an expanded review of the literature and discussion to enhance consensus among the sleep surgeons that consider surgical management in patients with snoring and OSA.

## 1. Introduction

Sleep-disordered breathing (SDB) is a series of disorders, including snoring and obstructive sleep apnea (OSA) [1]. According to the Merriam-Webster dictionary, snoring is defined as breathing during sleep with a rough, hoarse noise due to vibration of the soft palate [2]. This definition assumes that the source of snoring is the soft palate. Although there are other sites, the soft palate is the most common source. The vibration of enlarged tissues within the oro- and hypopharynx that causes snoring sounds results in inflammation. It is known that habitual snoring is a risk to developing OSA, which is a disorder with a high prevalence afflicting more than 100 million adults worldwide. Up to 90% of individuals with OSA remain without a diagnosis or therapy. Patients with untreated OSA are at increased risk for hypertension, cardiovascular disease, heart failure, stroke, obesity, metabolic dysregulation, diabetes mellitus, daytime sleepiness, depression, accidents, and all-cause mortality [3,4,5,6,7,8,9].

There are different treatment options for both snoring and OSA. The decision-making process to select a treatment is complex. It considers many factors, such as patient age, comorbidities, stage, severity of the disease, pheno- and endotypes of OSA, phenotypes of the palate, and anatomic features of the upper airway [10,11]. Results from additional diagnostic procedures, such as drug-induced sleep endoscopy, craniofacial computed tomography, and dynamic sleep magnetic resonance, are also considered [12,13].

It is common practice to discuss non-surgical treatments with snoring and OSA patients. These treatments include weight loss, sleep hygiene, the use of oral appliances, and undergoing myofunctional therapy [14,15,16,17,18]. Positive airway pressure therapy (PAP) represents the first-line treatment option for moderate to severe OSA under indications and is followed by PAP titration [19].

In cases where the above method fails, surgical procedures might be considered. They are divided into minimally invasive and invasive ones. It is critical to decide which patient would be suitable for surgical treatment and the steps and criteria for selecting a particular surgical option. Moreover, for those patients who are considered for surgery, it is essential to know the peri-operative considerations and decisions that need to be made beforehand regarding post-operative management.

There are considerable differences in knowledge, experience, healthcare customs, and standards among countries, healthcare systems, and institutions. However, the patient and the provider are interested in establishing common concepts and standards. Whilst it is difficult to reach a set of standards worldwide accepted to have a more similar healthcare quality, this goal is more accessible to achieve for one continent.

Therefore, we established a panel of otolaryngology/head and neck experts in the surgical treatment of sleep-related breathing disorders, such as snoring and OSA, to develop statements on decision-making regarding this treatment and peri-operative considerations regarding snoring and OSA in adults, with the aim of reaching a shared statement’s consensus. In the first phase, the expert panel explored the agreement/disagreement on definitions, patient criteria, and diagnosis of snoring and OSA [20]. In the second phase, our goal was to assess the level of agreement in decision-making and peri-operative considerations.

## 2. Materials and Methods

To establish consensus statements among European experts on diagnosing and treating snoring and obstructive sleep apnea (OSA), experts in the field were invited to a panel. As outlined in our previous work [20], a structured approach was implemented, incorporating the modified Delphi method to determine the level of consensus [20,21,22,23,24]. After a thorough literature review conducted by the first author, an initial draft of the consensus statements was created. These statements were systematically categorized into definitions, patient criteria, diagnosis, considerations for surgical intervention, pre-operative considerations, procedures related to palatal surgery, post-operative outcomes, monitoring, and potential complications.

After collective review and discussion among the core group of three colleagues (EO, ADV, CV), the initial set of statements was established. The foundational team identified esteemed otolaryngologists with expertise in sleep surgery for snoring and OSA management, particularly palatoplasty. Consequently, a panel comprising 8 experts was formed through the invitation. Panelists were provided with an Excel spreadsheet containing the statements. They were prompted to indicate their agreement or disagreement with each statement and to provide comments or suggest modifications where appropriate or necessary. After collecting all feedback, the results were aggregated, highlighting the count and percentage of panelists agreeing or disagreeing with each statement, and calculated.

A second-round iteration was initiated in alignment with the modified Delphi method, aiming for a minimum of 75% consensus. Adjustments were made to the statements based on panelists’ feedback, and alterations were distinctly marked. The revised document presented the degree of consensus numerically and as a percentage and included anonymized comments from the panelists. Each participant was given their prior responses, with the opportunity to modify their feedback, offer fresh comments, or propose further refinements to the statements. Customized second-round files, reflecting these elements, were disseminated to each panel member.

After collecting feedback from the second round, subsequent third and fourth rounds of statement evaluations were undertaken using a similar methodology. Finally, a conclusive verification statement file was generated, highlighting the responses representing the minority stance among the eight panelists.

A definitive set of statements was consolidated upon collecting feedback from the verification files. Subsequently, a summarized representation, detailing the consensus percentage for each statement, was disseminated to all panel members. This summary ensured clarity by presenting an aggregate view, deliberately omitting individual panelist stances.

The first author then articulated a strategy to draft the initial manuscript, spotlighting statements that delineated the definitions and diagnostic criteria for snoring and OSA. Following the endorsement of this plan, EO embarked on the drafting phase, meticulously crafting each segment of this manuscript. These segments were sequentially circulated among the panel members, ensuring each section was vetted iteratively.

Following an initial examination and necessary adjustments to each section, the detailed manuscript was assembled in accordance with the target journal’s guidelines. This unified version was subsequently distributed among the panel members, encouraging them to perform an in-depth evaluation, propose improvements, and provide their approval. Only after obtaining their consensus did the document move forward to submission.

## 3. Results

An initial set comprising 91 statements was disseminated to an expert panel to evaluate consensus regarding decision-making for the surgical treatment of snoring and OSA. This panel, comprising eight members, evaluated each statement and marked it either in agreement or in disagreement.

In the primary evaluation phase, the panelists registered their agreement or disagreement with an average of 90.4% of the disseminated statements. There was no response for 9.6% of the available statement options across all eight panelists.

A detailed examination of the entire initial set of statements indicated that 19.8% of the statements achieved unanimous consensus among all eight panelists agreeing, 14.3% secured the consensus of seven panelists, six panelists supported 12.1%, 13.2% agreed upon by five, and 7.7% agreed upon four panelists. Furthermore, 8.8% and 4.4% of the statements found consensus among three and two panelists. In comparison, 13.2% received the endorsement of just one panelist, and a mere 1.1% faced unanimous rejection, with none agreeing on that specific statement.

As described in the previous publication [20], the primary author analyzed the feedback, considering both the quantitative responses and the qualitative comments. This review developed a refined set of statements, which served as the second-round survey instrument. The revised document was organized to ensure clarity and precision, as described in Part 1. Definitions and Diagnosis [20]. Modified statements were marked, with summary metrics for each statement detailing the number of panelists who had responded and the count and percentage of those who agreed and disagreed with each statement.

As described in Part 1., anonymous comments for each statement were included. The second round included each author’s responses, and columns were added for any changes with additional comments and disseminated to each panel member.

Following the established research protocol, for both the third and fourth iterations, a comprehensive process of reviewing panelists’ feedback and quantifying the degree of consensus was conducted, and a renewed set of files was prepared and subsequently distributed to the panelists.

For the culmination of the study, a definitive round of verification files was contrived by drawing attention to the responses that diverged from the majority.

The terminal set of statements focused on the themes of decision-making and peri-operative considerations related to snoring and OSA, comprising 40 statements. A complex examination of this set reveals that 29 of these statements were autonomous, whilst a subset of 11 statements stood as overarching themes, each further bifurcating into 42 specific sub-statements. These sub-statements, delineated as “a”, “b”, “c”, and so forth, were intended to expound on the multifaceted aspects and intricacies of their respective parent statements. In summary, the final document on decision-making and peri-operative considerations included 71 autonomous sub-statements.

Within the segment delineated as decision-making in the surgical treatment of snoring and OSA (Table 1), of the 33 statements in the final round, a consensus was achieved among all eight panelists for 20 out of the 33 statements, equivalent to 60.6%. Moreover, seven out of eight panelists achieved consensus on 9 (27.3%) and six out of eight panelists agreed on 2 (6.1%) other statements out of a total of 33 statements. Overall, of the 33 statements, 31 (93.9%) demonstrated a consensus among at least six (75%) of the panelists (Figure 1).

In the statements of peri-operative considerations regarding the surgical treatment of snoring and OSA (Table 2), there were a total of 38 statements. Of these, 11 statements were stand-alone, and 6 parent statements had 27 sub-statements. A consensus was reached among all eight panel members on 21 out of 38 (55.3%) statements. On seven other statements (18.4%), there was agreement among the seven panelists. On an additional six statements (15.8%), there was agreement among the six panelists. Overall, of the 38 statements, 34 (89.5%) demonstrated a consensus among at least six (75%) of the panelists (Figure 2).

## 4. Discussion

There is a large variability in assessing and managing snoring and OSA. Patients with snoring and OSA may be in the practice territory of one or more disciplines. A patient seen by a physician performs the assessments, reaches a diagnosis, and typically offers the therapeutic options determined by that discipline’s perspective on these problems and their opinions on what works. Sleep surgeons are not immune to similar biases, favoring and offering surgical treatment options over medical, without severe review, consideration, or trial of non-invasive alternatives. As sleep surgeons, this expert panel finds it extremely important to explore a consensus on the decision-making process in selecting and offering treatment options to patients. Once surgical treatment is considered for a specific patient, decisions are needed regarding many peri-operative considerations for optimal patient care.

### 4.1. Delphi Method

The Delphi method and its modification secured the anonymity of individual panelists, avoiding the potential bias of dominance or group conformity, and the ease of changes in the proposed statements, when needed, to enhance the agreement between the panelists [20].

In the first round of the present study, 90.4% of the statements were answered, leaving 9.6% unanswered due to unclarity. Using the modified Delphi method, it was possible to eliminate misunderstandings and misinterpretations and have responses on all the final sets of statements.

### 4.2. Decision-Making—Patient Factors

In the section on decision-making in the surgical management of snoring and OSA, eight out of eight panelists reached a consensus in 20 out of 33 statements. There are essential statements that reached complete consensus among the panelists. Knowing the patient’s expectations and past medical history and discussing all treatment options, their risks, and expected outcomes is also necessary.

### 4.3. Decision-Making—Non-Surgical Options

While all panelists found a sleep study essential before the surgical treatment of OSA, seven out of eight recommended a sleep study before the non-surgical or surgical treatment of snoring. Most of the panel (87.5%) agreed to consider surgical treatment in patients with BMI > 40, specifically only in patients with Grade 4 tonsillar hypertrophy.

A trial of non-surgical options before surgery was recommended for both snoring and OSA, although the latter had an 87.5% consensus. There was a consensus on a trial of improvement of life quality measures such as diet, weight control, exercise, reduction in alcohol use, and recommendation of a repeat sleep study for those who showed reasonable compliance [25]. In a randomized study from 2006, Lam et al. showed that PAP produced the best improvement in terms of physiological, symptomatic, and health-related quality of life (HRQoL) measures, while the oral appliance was slightly less effective [26]. Weight loss, if achieved, resulted in an improvement in sleep parameters, but weight control alone was not uniformly effective. For patients with moderate to severe OSA, a sleep study with PAP titration was helpful by 75% but not necessary (87.5%) before the surgical treatment.

The panel concluded that a mandibular advancement device (MAD) trial before surgery was not found essential in all patients with OSA (one panelist dissented), not in patients with moderate OSA and patients with primary snoring, but could be considered in patients with mild OSA. The panel split 50:50 on the need for a trial with MAD before the surgical treatment. Lee et al. showed that sleep surgery and MAD are equally effective treatments for OSA, according to autonomic activity [27]. Determining optimal individualized treatment options for sleep apnea remains an area of ongoing research and debate among experts. Without evidence supporting personalized treatment for each patient, a step-by-step treatment protocol may provide better outcomes instead of leaving the decisions to individuals’ choice.

The impact of position on the presence and severity of OSA has been known for decades [28]. A positional treatment trial was found to be not essential in all patients with OSA among the panelists (one panelist dissented) or patients with mild or moderate OSA, but was found to be necessary in all patients with positional OSA [29].

### 4.4. Decision-Making—Positive Airway Pressure

The adequacy of the duration of PAP treatment has been debated in the literature. Considering most patients with OSA are only partially compliant with PAP, the efficacy and effectiveness of this treatment depend on the frequency and duration of its use [30]. An effective AHI index has been recommended to measure the effectiveness of PAP therapy based on the actual respiratory events while being on and off the machine [31]. A recently published review points to the optimal hours of PAP use [32]. The authors claim that the optimal hours of PAP use are an individualized definition. The previously recommended rigid hours for its use should be discarded. Clinical indicators could be improved by using PAP for an appropriate percentage of total sleep time. In our study, most of the panelists (75%) defined PAP compliance as adequate if it is used at least five nights per week and for at least 4 h per night.

In patients who are not compliant with PAP use, there was a consensus on first investigating and addressing the mask, PAP settings, and nasal obstruction-related problems. Regarding nasal obstruction, there was consensus on prioritizing medical and/or surgical management before surgical management.

### 4.5. Informing, Shared Decision-Making

The panel of experts reached a complete consensus on the need to inform the patient of all the non-surgical options before surgery for OSA. Also, after gathering knowledge of those treatment methods, the patient’s desire to proceed with the surgical treatment would be considered sufficient for surgical treatment. Similarly, there was consensus that the patient’s reporting of non-compliance with or desire not to try PAP was deemed sufficient (except for one dissent) for choosing surgical treatment; however, solely, the patient’s decision to have surgery was not considered sufficient to proceed with surgery.

The consideration of patient preference and adherence to treatment regimens is crucial in clinical practice. Studies have shown that patient-reported preferences and experiences significantly affect treatment decisions and outcomes. In the context of OSA treatment, patient non-compliance with or unwillingness to try PAP therapy may indicate a need to explore alternative non-CPAP therapy options, including surgical intervention. However, it is essential to acknowledge potential barriers to treatment adherence and address them through patient education, support, and personalized care plans. Furthermore, shared decision-making and surgical intervention emphasize collaboration between patients and healthcare providers in making treatment decisions. While patient preference is important, it should be integrated into a shared decision-making process that incorporates clinical evidence, expert recommendations, and individual patient values and circumstances. Research suggests that shared decision-making can lead to more informed choices, increased treatment satisfaction, and improved health outcomes. Therefore, while patient desire for surgery is a significant factor, it should be evaluated within the framework of shared decision-making to ensure that it aligns with the patient’s overall treatment goals and preferences.

There was complete consensus on the need for detailed documentation of all the elements of decision-making, patient expectations, and all the shared information.

### 4.6. Pre-Operative Workup

In the section for peri-operative considerations regarding the surgical treatment of snoring and OSA, the eight panelists reached a complete agreement in 21 out of 38 statements.

There is a lack of evidence in the literature regarding the standard pre-operative workup for the management of OSA. A study by Bamgabde et al. highlighted variations in anesthesiologists’ peri-operative care of OSA patients, even in developed countries with advanced medical training and standards [33]. This is generally reflected in our study’s failure to achieve a high level of consensus in all aspects of pre-operative workup. On the other hand, there was a high consensus regarding the need for a chest X-ray, EKG, and anesthesia consultation, but no need for an echocardiogram or cardiac consultation. Moreover, there was consensus on the need for all these elements above pre-operatively (except one dissent for the cardiac consultation). There was consensus that there was no need for anesthesia consultation in all patients with OSA. Still, it was considered required in all patients with severe OSA, cardiovascular co-morbidities, MOS < 90%, and BMI > 40. The level of consensus among the panelists is summarized in Table 3.

### 4.7. Surgical Setting Considerations

There was a consensus among the panelists on the fact that surgery for snoring and mild OSA (even if BMI is high, i.e., >28, with one dissent) could be performed in outpatient facilities. On the other hand, the panel reached a high level of consensus (87.5%) for the need to perform surgery in moderate to severe OSA in an inpatient facility and to keep the patients overnight. The panel had a 75% agreement on the statement on the need for all patients who receive general anesthesia to be kept overnight.

Regarding the need for performing the surgery in a facility with an ICU, the panel reached a complete agreement for the patients with cardiovascular co-morbidity, but not just for the saturation below 80%. There were split recommendations (50:50) regarding BMI > 35, prolonged apnea durations, and diabetes.

### 4.8. Monitoring Considerations

The panel reached a complete consensus regarding the need for continuous pulse-oximetry monitoring of the inpatients and the need for a specific pain management protocol to be established by a sleep center. Zhang et al. identified continuous pulse oximetry and capnography as keys for detecting apnea effects [34]. While photoplethysmography (PPG) accurately measures ventilation, oxygenation, and respiratory rate, its sensitivity needs improvement. Traditional clinical monitoring is inadequate for continuous tracking. Combining continuous pulse oximetry with a respiratory volume monitor could enhance patient safety, given their complementary functions, meriting further study but also showing the need for monitoring [20].

### 4.9. Degree of Consensus

The consensus among the experts was significantly greater in areas of agreement than in areas of disagreement. However, the decision was made to retain the latter statements to clearly indicate that there are still unresolved issues within OSA’s clinical and therapeutic dimensions, emphasizing the need for ongoing research and dialogue in these areas.

The authors believe that the statements in the current study on decision-making and peri-operative considerations regarding the surgical management of snoring and OSA will guide clinicians, particularly sleep surgeons, how to manage these patients. Continued discussion is expected to enhance standardization and stimulate further research in areas that lack consensus.

## 5. Limitations

This study has a number of limitations, starting with the small number of panelists. The Delphi method is widely used among experts for consensus-building but has its own set of challenges. One of the primary concerns is expert selection bias. This occurs when the selection of experts is not sufficiently diverse, leading to outcomes that might be skewed or overlook essential viewpoints. The risk is exacerbated if the panel includes experts with similar views or if some participants have conflicting interests or biases, which could undermine the credibility of the consensus reached. Another significant issue is expert attrition. Obviously, authors cannot and do not claim that they represent the continent of Europe. The participation of more panelists that represent each European country would have been much more difficult and time-consuming. The Delphi method typically requires multiple rounds of feedback and iteration, and maintaining consistent participation from experts throughout this process can be challenging. The risk of experts dropping out or losing interest as the process unfolds can distort the final results, making it difficult to achieve meaningful outcomes. Moreover, it would be unlikely to reach the level of consensus achieved in the current work. There would be a problem in setting the criteria for representation and selecting the panelists. The number of otolaryngologists who have sleep surgery as their primary practice field is limited overall; moreover, there are large differences in the availability of such sleep surgeons in each European country. The authors have prioritized the exploration and dissemination of levels of agreement among selected sleep surgery experts. This manuscript focuses on the decision-making related to surgical decision-making and is therefore relevant primarily to the surgeons that perform sleep surgery and may not be directly relevant to other specialties that participate in the assessment and management of snoring and OSA.

The second limitation was the need for multiple rounds of distribution of the statements. The success of the Delphi method greatly depends on receiving timely and substantial input from experts in each round. In each round, statements were changed as per the feedback received by one or more panelists, taking the risk of resulting in disagreement with the revision from the other panelists. Challenges such as low response rates or superficial feedback can significantly impede the consensus-building process. This makes it harder to reach a consensus and can also prolong the entire process, affecting the method’s overall effectiveness. A large number of statements ended up being modified multiple times.

The third limitation was the inability to have all members meet at the same time, in person or online, to discuss the statements that did not reach a full or high level of consensus. Although not required by the Delphi method, this could have increased the rate of consensus among the panelists. This modification of the Delphi method may be considered in future studies.

## 6. Conclusions

Assessment and management of snoring and OSA demonstrated significant differences in the world concerning the disciplines, concepts, methods, and techniques. On one hand, this highlights the complexity of these clinical presentations and the variability in training and resources. On the other hand, this manifests a lack of consensus among and within the involved disciplines. Before making progress on selecting a specific surgical treatment modality, consensus is essential in the decision-making process in the surgical management of snoring and OSA.

This consensus statement offers a summary of outcomes from a collaboration among European sleep surgery specialists. It aims to establish guidelines on decision-making and peri-operative considerations for snoring and OSA surgical management.

The diverse opinions among experts, especially on alternative treatment options before surgical consideration, highlight the complex nature of OSA. This diversity underscores the importance of identifying critical areas for future research and improvement in clinical practices. Additionally, it serves as a foundation for achieving agreement on treatment approaches and outcomes.

## Figures and Tables

**Figure 1 jcm-13-02083-f001:**
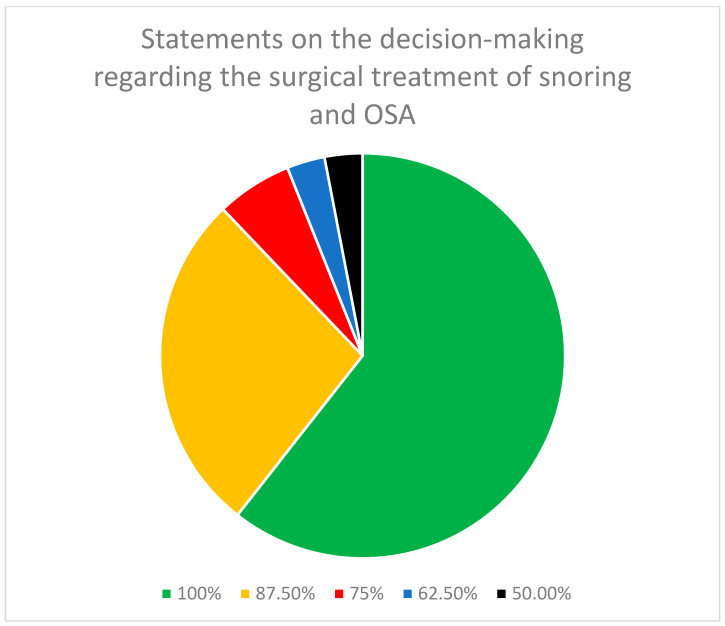
Distribution of the degree of consensus among panelists for the statements on decision-making regarding the surgical treatment of snoring and OSA.

**Figure 2 jcm-13-02083-f002:**
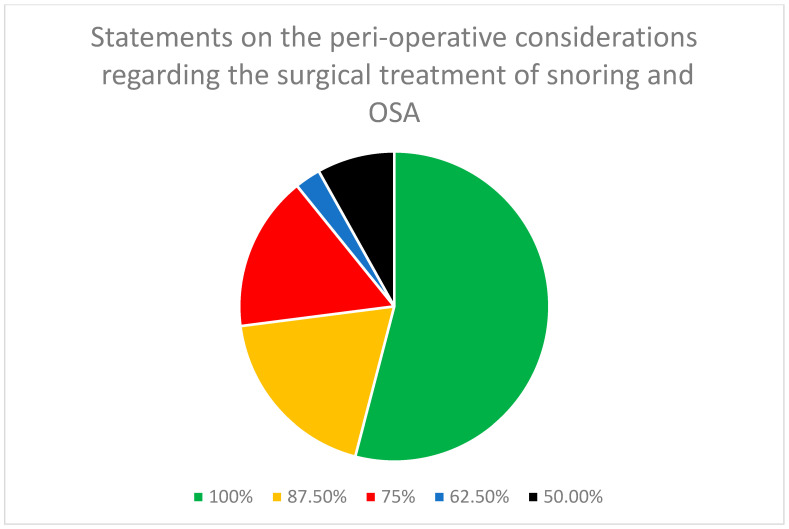
Distribution of the degree of consensus among panelists on the statements for the peri-operative considerations regarding the surgical treatment of snoring and OSA.

**Table 1 jcm-13-02083-t001:** Statements on decision-making regarding the surgical treatment of snoring and OSA.

Statements  100% 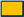 87.5% 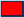 75% 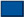 62.5% 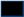 50%	% Consensus
1. Knowing the history of past diagnostic and treatment selections and outcomes for snoring and OSA is essential for decision-making in management.	 100
2. Learning the expectations of the patient regarding the management of the patient’s diagnosis.	 100
3. Before the non-surgical management of snoring, a sleep study is essential.	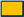 87.5
4. A sleep study is necessary before consideration for any surgical treatment for	
a. snoring	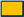 87.5
b. OSA	 100
5. For the management of snoring and OSA, discussion of all treatment options, their risks, and expected outcomes is essential.	 100
6. Surgical treatment may be considered in patients with a BMI > 40.	 100
7. Surgical treatment should not be considered in patients with a BMI > 40, with the exception of tonsillectomy only in patients with Grade 4 tonsillar hypertrophy.	 100
8. For the management of snoring and mild OSA, non-surgical treatment options should be recommended.	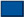 62.5
9. As a general rule, non-surgical options should be tried prior to surgical options for the treatment of	
a. snoring	 100
b. OSA	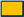 87.5
10. All patients should try improving their health quality measures (diet, weight control, exercise, reduce alcohol).	 100
11. Those patients who are enthusiastic and willing to improve their life quality/to address the risk factors should have repeat sleep studies after adequate trials.	 100
12. For the patients that are potential candidates for surgery in cases of moderate or severe OSA, a sleep study with PAP titration is	
a. useful	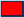 75
b. not necessary	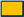 87.5
c. not essential	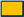 87.5
13. If there are no contraindications, before any surgical treatment, a mandibular advancement device (MAD) trial is	
a. not essential in all patients with OSA	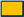 87.5
b. not essential in patients with primary snoring	 100
c. may be considered in patients with mild OSA	 100
d. not essential in patients with moderate OSA	 100
e. not essential to try before the surgical treatment	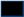 50
14. Before any surgical treatment, a positional treatment trial is	
a. not essential in all patients with OSA	 100
b. essential in patients with positional OSA	 100
c. not essential in mild OSA	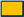 87.5
15. PAP compliance is considered adequate if it is used at least 5 nights per week for at least 4 h per night.	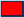 75
16. In patients with non-compliance, the following causes of non-compliance should be investigated and addressed: a. mask size/shape and mask fitting on the face; b. PAP settings; c. nasal obstruction, and if present, the cause of the nasal obstruction.	 100
17. In patients for whom nasal obstruction is the suspected cause of non-compliance with PAP, medical and/or surgical management of nasal obstruction should be priorities for helping PAP compliance (in patients motivated for PAP treatment) prior to the sleep surgery.	 100
18. Patients should be informed adequately of all non-surgical options before considering the surgical treatment of OSA.	 100
19. After adequate information is provided, the patient’s desire not to try any non-surgical treatment is sufficient to choose surgical options.	 100
20. In a patient who is adequately informed of the options and without a pulmonary condition, the patient’s desire not to try PAP treatment is sufficient to choose surgical options.	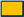 87.5
21. Patient reporting of non-compliance with PAP is sufficient to choose surgical options.	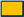 87.5
22. The patient’s desire to have surgery is not sufficient to proceed with the surgery.	 100
23. Detailed documentation of all the elements of decision-making, the patient’s expectations, and the information given by the surgeon is an essential part of the management of snoring/OSA.	 100

**Table 2 jcm-13-02083-t002:** Statements on the peri-operative considerations regarding the surgical treatment of snoring and OSA.

Statements  100% 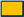 87.5% 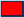 75% 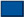 62.5% 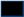 50%	% Consensus
1. For the pre-operative workup in all patients with OSA	
a. CBC, platelets, coagulation studies (PT, PTT), and EKG are required	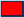 75
b. bleeding time/closure time are required	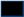 50
c. chest X-ray is not required	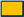 87.5
d. echocardiogram is not required	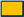 87.5
2. For the pre-operative workup in all patients with severe OSA	
a. chest X-ray is required	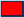 75
b. EKG is required	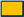 87.5
c. echocardiogram is not required	 100
d. disagreement with cardiac consultation is not required	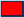 75
e. anesthesia consultation is required	 100
3. For the pre-operative workup in all patients with OSA and cardiovascular co-morbidities	
a. chest X-ray is required	 100
b. EKG is required	 100
c. echocardiogram is not required	 100
d. cardiac consultation is required	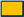 87.5
e. anesthesia consultation is required	 100
4. No particular age limit should be applied for pre-operative management.	 100
5. For the pre-operative workup in all patients with OSA and mean oxygen saturation (MOS) < 90%	
a. chest X-ray is required	 100
b. EKG is required	 100
c. echocardiogram is not required	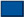 62.5
d. cardiac consultation is not required	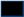 50
e. anesthesia consultation is required	 100
6. If surgery is indicated for the pre-operative workup in all patients with OSA and BMI > 40	
a. chest X-ray is required	 100
b. EKG is required	 100
c. echocardiogram is not required	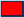 75
d. cardiac consultation is not required	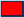 75
e. anesthesia consultation is required	 100
7. Surgery for snoring can be performed in outpatient facilities.	 100
8. Surgery for mild OSA can be performed in outpatient facilities.	 100
9. Surgery for snoring or mild OSA may be performed as an outpatient even if BMI is high (>28).	 100
10. Surgery for snoring or mild OSA may be performed under local anesthesia even if BMI is high (>28).	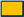 87.5
11. Surgery should be performed in inpatient facilities if the patient has moderate or severe OSA.	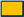 87.5
12. All patients that receive surgery should be kept overnight if they have moderate or severe OSA.	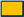 87.5
13. All patients with OSA who receive surgery under general anesthesia should be kept overnight.	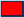 75
14. All patients with severe OSA should have surgery in facilities that have intensive care units	
a. If there is a BMI > 35 and/or prolonged apnea durations and/or diabetes	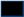 50
b. not necessary if there is a low saturation of <80	 100
c. if there is cardiovascular co-morbidity	 100
15. Inpatients should be monitored with pulse oximetry.	 100
16. Post-operative pain management is essential.	 100
17. All post-operative patients should receive pain management based on the protocol established by a particular sleep center.	 100

**Table 3 jcm-13-02083-t003:** Summary of consensus on recommendations of panelists regarding the pre-operative workup.

All Patients with the Following Conditions	Chest X-ray	EKG	Echocardiogram	Cardiac Consultation	Anesthesia Consultation
	Yes/No (% Consensus)				
OSA	Yes (75)	Yes (75)	No (87.5)	No (100)	No (100)
Severe OSA	Yes (75)	Yes (87.5)	Yes (87.5)	No (75)	Yes (100)
Cardiovascular co-morbidities	Yes (100)	Yes (100)	No (87.5)	Yes (87.5)	Yes (100)
Mean oxygen saturation (MOS) < 90%	Yes (100)	Yes (100)	No (62.5)	No (50)	Yes (100%)
Body mass index (BMI) > 40	Yes (100%)	Yes (100%)	No (75%)	No (75%)	Yes (100)

## Data Availability

Data are contained within this article.

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
