# Peer review of "Consensus Statements among European Sleep Surgery Experts on Snoring and Obstructive Sleep Apnea: Part 2 Decision-Making in Surgical Management and Peri-Operative Considerations"

_jcm, 2024, doi:10.3390/jcm13072083_

Round 1
Reviewer 1 Report
Comments and Suggestions for Authors
Overall, the manuscript offers valuable insights into decision-making in surgical management and perioperative considerations for snoring and obstructive sleep apnea. The detailed process of consensus building among the expert panelists is commendable. However, to enhance the document's clarity, I recommend minor revisions to improve the organization of the statements throughout the text.
Specific Comments:
- Improvement of Statement Organization: I suggest reorganizing the statements related to decision-making and peri-operative considerations to achieve better flow and coherence. Grouping related statements together under subgroup titles can significantly enhance the document’s readability.
- Addressing Limitations: Including a discussion section addressing the study's limitations would be beneficial. Describing this aspect of the research process is appreciated and contributes to the overall thoroughness of the manuscript.
Author Response
Authors appreciate the constructive comments and suggestions to improve our manuscript. Please find below, point by point responses to your comments.
Overall, the manuscript offers valuable insights into decision-making in surgical management and perioperative considerations for snoring and obstructive sleep apnea. The detailed process of consensus building among the expert panelists is commendable. However, to enhance the document's clarity, I recommend minor revisions to improve the organization of the statements throughout the text.
Authors thank the Reviewer for supportive comments.
Improvement of Statement Organization: I suggest reorganizing the statements related to decision-making and peri-operative considerations to achieve better flow and coherence. Grouping related statements together under subgroup titles can significantly enhance the document’s readability.
We understand that the Reviewer is referring to the discussion section. We have added subsections to the Discussion to enhance the document’s readability.
Addressing Limitations: Including a discussion section addressing the study's limitations would be beneficial. Describing this aspect of the research process is appreciated and contributes to the overall thoroughness of the manuscript.
We have added a section to the manuscript before the Conclusions and discussed the limitations of this current study.

Reviewer 2 Report
Comments and Suggestions for Authors
The authors developed a consensus on decision-making in surgical management and peri-operative considerations on snoring and OSA. A set of statements was developed based on the literature and circulated among 8-panel members of European experts, utilizing the Delphi method. The authors conclude that there is a need for an expanded review of the literature and discussion to enhance consensus among the sleep surgeons that consider surgical management in patients with snoring and OSA.
Overall, the study is well-conducted and I've some minor comments.
1. The authors should better describe the expert pannel: "experts in the field" is a rather limited affirmation. If the pannel of experts is constituted of surgeons only, one may doubt their ability to evaluate some of the questions such as: " For the patients that are potential candidates for surgery IN CASE OF MODERATE OR SEVERE OSA, a sleep study with PAP titration is ..."
2. When looking at their first publication (ref 18), it emphasizes that even for quite simple statements on the definitions regarding snoring and OSA, 100% consensus was not achieved (definition of obstructive sleep apnea, non-REM definition), which raises doubts about the expert nature of certain evaluators
3. In the discussion section, the authors state "One of the primary concerns is expert selection bias. This occurs when the selection of experts is not sufficiently diverse... " This statement is true, thus the pannel of experts should be better described. If only surgeons constitute the pannel, it is a limitation that should be acknowledged.
Author Response
Authors appreciate the constructive comments and suggestions to improve our manuscript. Please find below, point-by-point responses to your comments.
The authors developed a consensus on decision-making in surgical management and peri-operative considerations on snoring and OSA. A set of statements was developed based on the literature and circulated among 8-panel members of European experts, utilizing the Delphi method. The authors conclude that there is a need for an expanded review of the literature and discussion to enhance consensus among the sleep surgeons that consider surgical management in patients with snoring and OSA.
Authors appreciate the Reviewer’s supportive comments.
Overall, the study is well-conducted and I've some minor comments.
The authors should better describe the expert pannel: "experts in the field" is a rather limited affirmation. If the pannel of experts is constituted of surgeons only, one may doubt their ability to evaluate some of the questions such as: " For the patients that are potential candidates for surgery IN CASE OF MODERATE OR SEVERE OSA, a sleep study with PAP titration is ..."
We have stated in the manuscript that all panelists were otolaryngologists and sleep surgeons. We had a detailed discussion in the Limitations section regarding the selection of panelists. We added in the Limitations section the fact that all panelists were sleep surgeons, which creates potential bias. All physicians who are involved in the assessment and management of snoring and OSA see the issues from their perspective. This work did not have the aim of creating a multi-disciplinary panel and establishing consensus between different physicians. That is a different and not quite easy task. However, when a decision is made after failed PAP treatment or other non-surgical approaches, surgeons do consider surgical options and go through “decision-making” and “assessment of “peri-operative” issues, which is essentially within their field, involving other disciplines like anesthesiology or cardiology as it has been discussed in the manuscript. We believe that this task and the resulting content are clear to the reader.
When looking at their first publication (ref 18), it emphasizes that even for quite simple statements on the definitions regarding snoring and OSA, 100% consensus was not achieved (definition of obstructive sleep apnea, non-REM definition), which raises doubts about the expert nature of certain evaluators.
The Reviewer here criticizes the previously published manuscript. This is not relevant with respect to the current manuscript. However, the Reviewer implies that a dissent on one statement proves that the panelists are not ‘experts”. Perhaps concluding that this manuscript is also written by not experts in the field. Reviewer knows who the authors are, with their credentials and publications. the Reviewer has a specific negative rating about any of the authors, she/he can comment. Not that we will address this by changing any author, or defending the credentials.
If it was a suitable venue, we could get into interaction with the Reviewer regarding her/his opinions about consensus statements and how we reached and the dissents. But we all need to put out our opinions in manuscripts and publish them in journals like this.
In the discussion section, the authors state "One of the primary concerns is expert selection bias. This occurs when the selection of experts is not sufficiently diverse... " This statement is true, thus the pannel of experts should be better described. If only surgeons constitute the pannel, it is a limitation that should be acknowledged.
We did acknowledge and expand on this limitation in the relevant section.

Reviewer 3 Report
Comments and Suggestions for Authors
I would like to congratulate authors for addressing the important issue of treatment standardisation in snoring and OSA and for managing to elaborate a comprehensive and targeted consensus statement. Here are a few suggestions on my behalf:
In ‘Introduction’, some of the articles cited by the authors date before year 2000 and I would suggest maintaining more up to date references. Moreover, the citation should be more precise, for example for sentence ‘It is known that habitual snoring is a risk to developing OSA, which is a disorder with a high prevalence afflicting more than 100 million adults worldwide ‘the reference is not clear .
In ‘Materials and methods’, it might be useful to detail the criteria for choosing the expert panel.
In Results, there are no details regarding the comments made by the expert panel and how these comments further improved the final results. At least the most important comments and suggestions from the expert panel could improve the clarity of this consensus.
My recommendation for Discussions section would be to avoid repetition. It already is a fairly lengthy article, so in order to maintain the reader’s attention it would be best to keep it shorter. For example,
‘As discussed in our previous earlier work [18], the modified Delphi method was used to collect panel members' collective opinions. Delphi's method and its modification se-cured the anonymity of individual panelists, avoiding the potential bias of dominance or group conformity and the ease of changes in the proposed statements, when needed, to enhance the agreement between the panelists [18]. ‘ Another possible suggestion for maintaining a more appealing article length for the readers would be to eliminate the repetition of information from Table 2 regarding preoperative work-up, since there is an even better structured exposure in Table 3.
Author Response
Authors appreciate the constructive comments and suggestions to improve our manuscript. Please find below, point-by-point responses to your comments.
I would like to congratulate authors for addressing the important issue of treatment standardisation in snoring and OSA and for managing to elaborate a comprehensive and targeted consensus statement. Here are a few suggestions on my behalf:
Authors appreciate the Reviewer’s supportive comments.
In ‘Introduction’, some of the articles cited by the authors date before year 2000 and I would suggest maintaining more up to date references. Moreover, the citation should be more precise, for example for sentence ‘It is known that habitual snoring is a risk to developing OSA, which is a disorder with a high prevalence afflicting more than 100 million adults worldwide ‘the reference is not clear.
We have removed older references and added new additional relevant references.
In ‘Materials and methods’, it might be useful to detail the criteria for choosing the expert panel.
We have added information regarding the establishment of the expert panel in the material and methods section.
In Results, there are no details regarding the comments made by the expert panel and how these comments further improved the final results. At least the most important comments and suggestions from the expert panel could improve the clarity of this consensus.
The authors appreciate the comment and recommendation. We had outlined the method of developing and revising the statements after each round, in the methods section of the current manuscript, and we had referenced the earlier article “Reference #20”. Moreover, we have discussed the process and challenges in the “Limitations” section, which provides a better insight to the reader.
My recommendation for Discussions section would be to avoid repetition. It already is a fairly lengthy article, so in order to maintain the reader’s attention it would be best to keep it shorter. For example, ‘As discussed in our previous earlier work [18], the modified Delphi method was used to collect panel members' collective opinions. Delphi's method and its modification se-cured the anonymity of individual panelists, avoiding the potential bias of dominance or group conformity and the ease of changes in the proposed statements, when needed, to enhance the agreement between the panelists [18]. ‘
Repetitive sections, as we perceived, were removed or modified. A few paragraphs regarding the Delphi method were modified and incorporated in the “Limitations” section.
Another possible suggestion for maintaining a more appealing article length for the readers would be to eliminate the repetition of information from Table 2 regarding preoperative work-up, since there is an even better structured exposure in Table 3.
The authors appreciate the comments and recommendations. Table 2 has the same structure as Table 1, which is similar to the earlier publication, and planned third publication regarding palatoplasty. This structure not only provides a full set of results on a subsection of statements, but also, color codes with the associated figure more clearly demonstrate the distribution of level of consensus on that subset of statements. Therefore, we would prefer to keep Table 2 as is, not removing the repetitive statements that are covered in Table 3. Moreover, Table 3 is quite abbreviated, to make it easy for the reader to differentiate the conditions and levels of consensus. The authors don’t believe that the Reviewer suggests removing Table 3. This complimentary Table is therefore included in the Discussion section, while the structure of the presentation of results was kept uniform